# A Pattern of Collaborative Networking for Enhancing Sustainability of Smart Cities

**Corina M. Rădulescu [1,\*]**, **Svitlana Slava [2,\*]**, **Adrian T. Rădulescu [3]**, **Rita Toader [1]**,
**Diana-Cezara Toader [4]** and **Grațiela Dana Boca [1]**

[1] Faculty of Sciences, Department of Economics and Physics, Technical University Cluj-Napoca, 76 Victoriei Street, 430122 Baia-Mare, Romania; rita.toader@cunbm.utcluj.ro (R.T.); gratiela.boca@cunbm.utcluj.ro (G.D.B.)

[2] Faculty of Economics, Economics and Entrepreneurship Department, Uzhhorod National University, 3 Narodna Square, 88000 Uzhhorod, Transcarpathian region, Ukraine

[3] Faculty of Civil Engineering, Technical University of Cluj-Napoca,72 Observatorului Street, 400641 Cluj-Napoca, Romania; adrian.radulescu@mtc.utcluj.ro

[4] Faculty of Business and Economics (HEC Lausanne), Graduate School, Quartier de Chamberonne, The University of Lausanne, CH-1015 Lausanne, Switzerland; diana.cezara@gmail.com

\* Correspondence: corina.radulescu@cunbm.utcluj.ro (C.M.R.); svitlana.slava@uzhnu.edu.ua (S.S.)

**Abstract:** This paper represents a research response to the current vision on transformations regarding the capacity building of smart cities focused towards sustainability, by addressing the knowledge based urban development and collaborative tools that support the development, dissemination, and use of knowledge. The purpose of this paper is to develop a collaborative pattern of knowledge networking, focusing on sustainability goals within a smart city concept, using the logic of the Complex Adaptive System (CAS). The study was carried out in an innovation cluster in Romania; the Social Network Analysis (SNA) was used as a tool to perform the study. The results of this analysis, due to the suggested networking, have led to delimitation of the roles that Groups of Competences play to enhance the sustainability of smart cities in areas where the use of knowledge has the greatest impact. Results show that the success of the smart solutions' implementation depends on how the social and competence structures of the network are shaped and whether it permanently adapts to fit the sustainability objectives in the considered areas.

**Keywords:** complex adaptive system; collaborative smart city model; groups of competence; knowledge and innovation networking

## 1. Introduction

The smart city idea has been evolving over the last few years, witnessed by a bunch of research devoted to definition analysis of smart city concepts [1–7]. In their works it is highlighted that, today, a smart city concept is considered both as a theoretical approach (a gradual movement from the focus on IT modernisation of infrastructure in the late 1980s to a holistic approach of city sustainability in two decades) and as practical precedent in different formats [8], indicating those cases already well implemented across the world and in many places with smart city strategies. Digital technologies and networks are among the main fundamentals for smart city development [1,7,8]. Increasingly sophisticated and rapid changes in technology and social media have led to unprecedented structural changes, especially in urban areas, because the traditional methods and approaches cannot provide such effective and viable solutions. It therefore requires consideration of intervention mechanisms to support fair and sustainable community progress, using innovation systems. In the relevant literature,

particular attention is given to the question of increasing viability through innovation systems [9,10]; but the issues on how to improve the capability to structure the city's sustainability format [11] and the competence flow in the smart city model remain fragmented. In an attempt to fill this gap, this paper suggests developing a smart city concept based on a collaborative research network, which is leveraged by emerging sustainability goals, rather than separate work in selected sustainability directions within a smart city frame.

The development of a smart city is becoming an increasingly complex challenge given the changed that has appeared in the urban environment. Rapid changes in the priorities and perspectives of cities necessitate an integrated approach toward information, knowledge, technologies, and processes that have to be articulated in a systemic vision. In this context, urban planners are launching more and more projects based on technologies to respond to the challenges of city expansion and finding the proper infrastructural solutions. This means that technology, research, and innovation should no longer be a target, but become a core component of the smart city, forming its adaptability to emergent changes [1,4,12].

A greater understanding of adaptability is given by the concept of Knowledge Based Urban Development (KBUD). It suggests that a city "that learns", a city "based on knowledge", forms the demanded intelligence for practicing in different smart dimensions: transport/mobility, housing, safety, health, technology, infrastructure, governance, and education [13,14]. In the process of creating a smart city frame, there is a need for cohesion, convergence, and consensus in the dynamic management of the cognitive processes of creating, integrating, and implementing the knowledge and innovative ideas that will lead to better community sustainability. These processes generate the continuous transformations in partnerships and networking related to the hottest issues of city sustainability, that together can manage and provide more viable and applicable solutions [4]. From this point-of-view, universities are seen as important catalysts in the innovation system of smart cities, with a particularly conducive environment for generating new ideas, solutions, and strategies [15,16].

Along with this, there are a number of researches proving that collaboration is a necessary base for smart city practical implementation [7,17–19]. The main message of their elaborations focuses on collaboration, innovation, social networks, and IT. Some works are devoted only to organizational structures, while others [1,7] are also deeply involved in digitally based smart city modelling.

In sum, the analysed literature sources allow us to conclude that there is currently great emphasis on collaborative models for sustainable smart city development. It relates to both aspects: (1) a way to balance the three E's (economy, equity, and environment) of sustainability by agreement of the priorities among the certain community bodies and groups), and (2) actually making a city smart through the use of edgy knowledge, technology, and communication means, to solve the different aspects of the city's sustainability. It is quite clear that knowledge-based structures possess the biggest portion of the necessary expertise to foster both sustainability and smartness in a community; involving them in adaptive networking would increase their potential collaborative power.

However, despite this interest in a collaborative concept of a smart, sustainable city, no research was found that suggested how to better structure the local knowledge profile into competence groups and how to connect them to community sustainability goals. In this research, the logic of the Complex Adaptive System as an overall base for a smart city concept and Social Network Analysis as a technique for analysis were used to complete these research objectives.

Thus, the aim of the paper is to substantiate a collaborative pattern of knowledge networking with focusing on sustainability goals within a smart city concept, using the Complex Adaptive System (CAS) logic. In this way, the internal strengths are used for moving towards community sustainability. According to [7], the paper covers two of the three layers of a smart city for analysis: (1) the city layer (the factors associated with knowledge-intensive activities and knowledge infrastructure); and (2) the information and knowledge layer (institutional settings for knowledge flows and cooperation in technology and innovation).

For achieving this goal, the study was focused on the following research questions:

(a)　　What theoretical basics fit sufficiently for defining a smart city frame, capable of solving complexity and adaptability issues of sustainability?

(b)　　How does one configure a collaborative networking model, which integrates the CAS characteristics of the smart city?

(c)　　How can the connections and relationships of a network be established empirically, so as to generate a flow of knowledge, ideas, and solutions of excellence for a smart city?

(d)　　What are the benefits of configuring Groups of Competences within an innovative cluster, by means of Social Network Analysis (SNA)?

In this research, the Complex Adaptive System was used as the main approach for understanding complexity and adaptability of the smart city dynamic with a holistic sustainability frame; and Social Network Analysis served as a tool for mapping and understanding the knowledge flow between different knowledge-based organizations within a considered area. SNA was particularly helpful in identifying the major elements of influence in a structured smart city collaborative model of applied innovative decisions, in order to better and more efficiently provide diffusion of those innovations that are focused on the proper sustainable goals of a smart city.

The next part of the paper is organised in the following way. The second part discusses complexity and adaptability in a collaborative smart-city model with special attention given to innovative networks, serving as the adaptive agents of the responses to sustainability issues; it also suggests a conceptualizing frame of the Complex Adaptive System (CAS) for shaping a smart city network of competence groups, focused on sustainability goals. The third part describes the methods used for defining collaborative competence groups in the regional innovation cluster Cluj ICT, operating the Social Network Analysis (SNA) as an analytical tool, and provides a research justification for them. In the fourth part, the interpretation of the results of the formed networks of competence groups is given. The last part is devoted to the conclusions.

## 2. Theoretical Frame and Literature Review

### 2.1. Complexity and Adaptability in a Collaborative Smart City Model

A number of literature sources suggested that cities were always defined as complex systems [19], and thus it was not possible rationally to reduce it to a variety of dimensions, even if it was considered to be the best fitting one. The concept of a smart and sustainability-focused city is an even more complex notion [2,20–22]. The term "smart city" came after some time of exploitation of the general sustainability approach, which is now well-known as the balance of the three E's (economy, equity, and environment). However, its move towards sustainability suffered from several key issues, covering low community awareness in sustainability values, local context, general geographical factors [11,21], disintegrated management [2,21], and a lack of/and consequently a need for technologies, supporting implementation of accepted sustainability decisions [7,21]. Thus, a smart city concept was actively used during recent decades, especially the last five years [3], demonstrating different and efficient ways of involving digital technologies for more sustainable solutions [22]. The other study [23] reveals new debates on smart cities, focused on the four dichotomies concerning strategic principles that support smart city development: (a) Technology-led; Holistic; (b) Top-down; Bottom-up; (c) Double Helix; Quadruple Helix; and (d) Mono-dimensional; Integrated intervention logic.

Even now, there is no clear and common definition of the term "smart city". Some authors generalized [8] two approaches for a definition analysis:

(1)　　mono-topic descriptions that accent on one side of urban development—ecology, technology, etc.;

(2)　　with emphasis on interconnections of different (better—all) aspects of urban life, based on "multi-stakeholder, municipality-based partnership".

Our paper idea is based on the second approach, proving the importance of a holistic view on sustainability and smart cities (in a separate or a mixed formats), which is supported by extensive research [5–7,11,24,25] and many other studies that have appeared during recent years.

An analysis of the literature reviewed allows us to conclude that smart city models are ones that (1) provide a certain path to sustainability, (2) are founded on edgy knowledge, innovations, and digital technologies, and (3) use special collaborative networks, based on a communication grid. A possible model could be a multi-stakeholder partnership within a municipality aimed at addressing common interests and problems, using ICTs that underpin a "Smart classification" (Smart Governance, Smart People, Smart Living, Smart Mobility, Smart Economy, and Smart Environment [25]).

Thus, effective and rapid, collaborative decisions in knowledge-based networks are the agents of adaptability and tools for designing better, smarter paths to sustainable cities. In our research, knowledge-based networks are defined as ones formed of knowledge-generating and knowledge–acquiring organizations (producing innovations).

Knowledge-generating organizations include university and research organizations/divisions, which produce knowledge applicable for sustainability goals within the smart city concept. That means, in such networks knowledge exchange will happen between knowledge-generating organizations and different sectors of the community; thus, any type of organization—municipal, social, or business.

Sharing technological knowledge with multiple agents who are dispersed not only geographically, substantially hinders innovation capabilities. Although the organizational forms of knowledge-based/innovation networks can evolve, the risk of dispersion of knowledge or absence of knowledge-based companies can lead to failures of innovation systems, due to a lack of information within an innovation network [26]. A good network, based on interorganizational collaboration, can bind and enhance through the skills of all partners [27,28]. Extensive collaboration between organizations could take a form of multi or bilateral partnerships [29–31]. Participating in collaborative innovative networks within the smart city frame, they serve both the community sustainability goals and, through acceptable innovations, better survival in the market, if this is a business type of partner. Thus, on a higher level, we see the benefits of diversity and complementarity of the network together with the intense connections between the activities involved.

The smart city model operating with different issues will require a knowledge cooperation of different sectors, and universities or the university's structures could become centres (nodes), generating knowledge in or attracting it from outside smart city networks. The universities could serve this purpose much more effectively and play their special role in the community, not only by teaching, but also by accompany innovation in a municipal area, being the generator of smart change, organizing a reflective learning path, and addressing the hottest community problems and required developments [32]. Thus, a smart city is a model that creates a learning city format as a parallel to learning regions; the latter are described by a number of authors [33]. The knowledge space has to be formed through collaboration in learning between city partners and thus creating the innovative information–delivery networks, supported by innovation policy for creativity, inspiration, and for reflection [32,34,35].

The information exchanges in innovative networks, impacting the acceleration of sustainability solutions, depend heavily on social capital development, and more on a bridging than on a bonding concept of social capital. Because innovation is formed also as a combination of knowledge, it is worth having relations outside of the network [36]. More support to this is given by the concept of "post-learning region" that stresses learning as being a main feature of the dynamic of social capital [37,38].

Thus, we suggest that a smart city collaborative model is knowledge reflective and a learning city model, generating permanent support to open knowledge-based/innovation networks of competence groups, with a high focus on sustainability, creativity, proper cultural context, and an effective governance model.

*2.2. Complex Adaptive System (CAS) as a Conceptualizing Frame for Shaping a Smart City Network of Competence Groups*

General theoretical research in more recent studies described the main principals required to conceptualize Complex Adaptive Systems [38–42], and many others] using three main layers of CAS features: system, complexity and adaptability (see more details in Table 1).

**Table 1.** The use of Complex Adaptive System (CAS) components for conceptualizing smart-city sustainability network of competence groups.

| CAS Components * | Analyzed Objects or Processes (based on the Case Study) | Results | Used Techniques |
|---|---|---|---|
| *Understanding system* | | | |
| Agents | Competence objects | | |
| Interdependence/Autonomy | Level of agents' autonomy | | |
| Emergence | Appearance of new elements | (1) Local competence profile | Survey, statistical analysis |
| Connectivity | Organizations' links | | |
| Feedback | Nature of an agents' feedback to a system | | |
| *Understanding complexity* | | | |
| Diversity and Modularity | Level of commonality/ differences between competence objects | | |
| Uncertainty, Distributed Control | Level of system control, if any | | |
| Self-organization | Increasing the capacity of joint agents' work instead of control | (2) Competence groups, focused on sustainability goals | Social network analysis |
| Adaptation | The agents' and system's learning | | |
| Transformation | Challenging sustainability goals and institutional reconfiguration of agents towards the problem solving. | | |
| *Understanding adaptability* | | | |
| Far from equilibrium | Floating balance of sustainability goals and agents' interests | | |
| Space of possibilities, new niches, prediction of future | Sustainable initiatives generated by competence groups and a system in a whole | | |
| Co-evolution | Joint agents' work on sustainability goals implementations | (3) Innovations as smart solutions towards sustainability goals through adaptive networking of clustered competence groups | Social network analysis |
| Historicity and time path-dependence | Dependence on the previous research | | |
| Networking | Possibility of daily-based forums of competence groups | | |
| Creation of new order | Innovative cluster of competence groups | | |

* Completed by authors with a modified grouping of the CAS components in selected literature.

The fundamental definition of complex adaptive system states that explanation of separate parts does not directly cause a perfect interpretation of behaviour of a whole system [43]. Exploring this definition we can easily highlight the main features of evidence of complexity and adoption as (a) there are many different agents, usually covering the interdisciplinary parts in the system, (b) they are not static, but dynamic (behavioural), and (c) the system endogenous process through the parts' dynamics forms the whole system's behaviour, which is distinctive from a sum of actions of the parts involved.

Carmichael and Hadzikadi [42] stress the usefulness of organized versus non-organized complexity; the first one leads to a system with a large number of heterogeneous agents with correlated interactions, which are able to organize the natural, endogenous feedback flow.

Thus, a concept of a sustainable city as a smart-organization can be considered as a complex adaptive system, which is functioning on a base of permanent adaptations of human actions towards the desired sustainable state, resulting from a city's behaviour as a whole system, consisting of natural, economic, social, and other involved components. The networked groups of competence, as an adaptive part of a smart city, will behave in the same manner (Table 1).

An adaptation happens when the parts involved in a system react on internal and external changes, pushing a system as a whole to become self-organized in the new quality of dynamic networking to together solve an appearing problem or to use the emerging possibilities towards sustainability. Thus, the network of Groups of Competence through holistic intelligent reaction derived from the features of CAS, allow getting the precious system power, which is logically greater than simple "aggregations of the individual static entities", even if they could be comprised of "very simple agents" [42–44].

Back to the period of two decades ago, Pumain [45] also considered a city evolution (city transformation) as an adaptive process and described cities as open systems with adaptive structures and the "power of creativity".

In our research the Complex Adaptive System theory (CAS) was used (following Carmichael and Hadzikadi, [42]) as a frame for understanding a network of agents that collaboratively could form a smart-city shape and respond to sustainability goals. The sustainable move will depend on how much the different components are driven by common sustainability goals.

The Collaborative Smart City model (Figure 1) developed in our research, analyses the Groups of Competences as agents of change, having a high level of autonomy of the members who quickly interact with each other, through an exchange of ideas and solutions, and offering a prompt feed-back to the system. This is an on-going process, proactive to changes in the external environment.

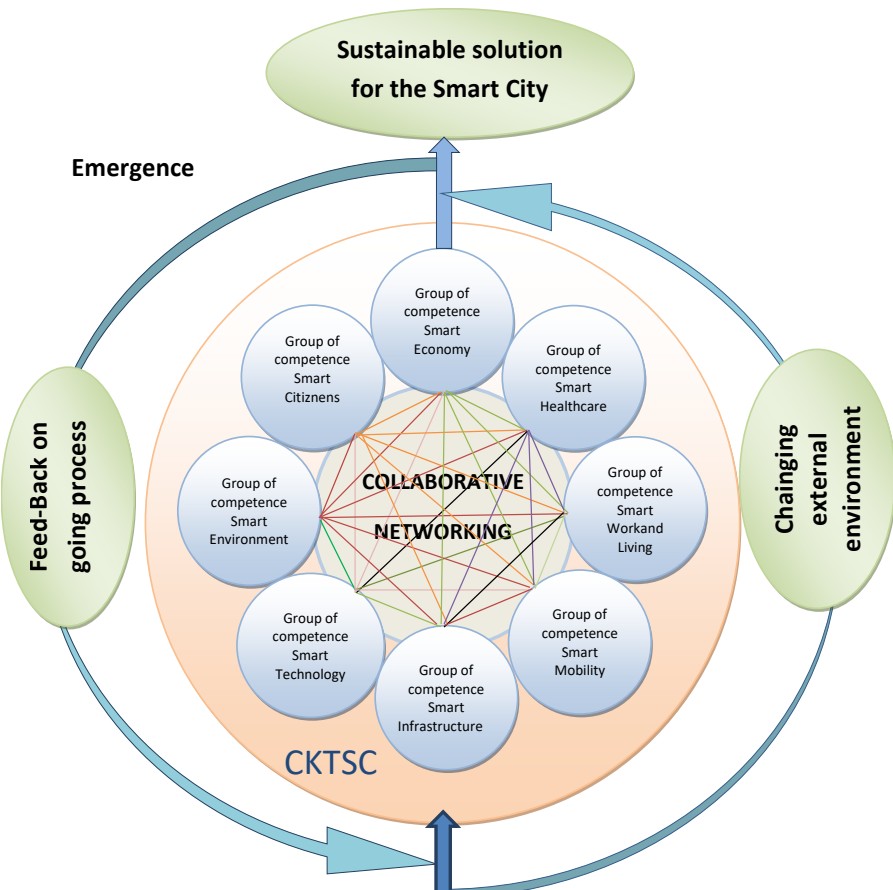

**Figure 1.** The Smart City Model. Source: the authors (CKTSC: Centre of Knowledge Transfer for Smart Cities).

The complexity of the collaborative model is manifested by the heterogeneity and inter-disciplinarity of the Groups of Competences. Their members are researchers from different fields, grouped by their competences on the main dimensions of development of the smart city: Smart Infrastructure, Smart Technology, Smart Mobility, Smart Work and Living, Smart Healthcare, Smart Citizens, Smart Economy, and Smart Environment [22]. They work together in high interconnectivity, having in common a goal towards excellence in innovative and smart solutions through the diffusion of knowledge. In this model, collaborative networking is aimed at solving problems related to improving the sustainability of smart cities.

In terms of adaptability, the model meets the smart city objectives without compromising on achieving the qualitative sustainability objectives imposed by the competent groups (agents).

## 3. Methods

### 3.1. Research Approach

This applied research is oriented towards understanding the complex problems arising in different environments [46–49]. Qualitative and quantitative research provides an effective support for this type of research. The investigation of complex and adaptable environments, such as smart cities, is successfully carried out when using studies [50]. These are relevant, especially in the context of researching networks, where the rapid transmission of information and knowledge flows play a very important role in the management of the complex system [51,52]. As a research method, we used SNA as a useful tool in the investigation of organizational structures regarding collaborative structures based on networking. Thus, social network analysis is an integrated design and strategy method, both for supporting the processes of building inter-organizational groups, and for communicating and exchanging organizational knowledge [53,54]. Using social network analysis in the research of innovation processes is mainly reasoned by the need to explain and describe the causal social mechanisms related to innovation.

### 3.2. A Social Network Analysis of Groups of Competence in the Innovation Cluster-Cluj ICT

Due to the lack of adequate empirical evidence regarding the contributions of actors from a knowledge-based smart city structure, we conducted a study on an innovation cluster from Romania in which the collaborative structures (Groups of Competences) were analysed by the SNA method. To conduct this analysis and to support the research hypotheses, we used as a conceptual model, the collaborative networking projected in the previous section. The organizational model suggests that the centres of competence are elements that provide a knowledge base and research excellency for the focus and development of cities towards innovative and intelligent systems. The effects of the interactions of the flows of the system of cities based on knowledge represented by its basic components Economy, Society, and Environment, generate dynamic actions that, by using the knowledge, can trigger the emergence of innovation systems. In addition, information and communication technology is an incentive to create innovative smart systems, which adds value to existing ones.

### 3.2.1. Innovation Cluster Cluj ICT—Overview

The innovation ITC Cluster was established on the initiative of the North-West Regional Development Agency, as a catalyst organization. The clotting of member entities and the construction started about seven years ago. The innovation Cluj ITC cluster is a sophisticated organization that is active in information technology. It has in its composition 34 companies providing software services and solutions, 5 academic institutions, 2 public bodies, and 7 other catalyst organizations.

The Cluj ITC cluster falls into the category of innovative clusters through its history, aspirations, and concerns in R&D, with the intent to create a centre for research and technology transfer and achieving significant investments in R&D.

The cluster aims to create an ecosystem conducive to the development and marketing of innovative software services and products, with high added value, through cooperation, exchange of knowledge and ideas, public–private partnerships, and supporting research and innovation, all for the benefit of member organizations. Thus, member entities have joined efforts to respond more effectively in terms of the level of human resources, R&D and innovation, business infrastructure, funding, marketing and sales, in order to make the transition to a higher level of design, development and marketing of software solutions [55]. Our proposal is to advise the CEO of the Cluj ITC Cluster to establish the Centre of Knowledge Transfer for a Smart City, which is a research structure, built as a network of groups of researchers from the universities' cluster members. The Centre of Knowledge Transfer for a Smart City will be one of the strategic directions adopted by Cluj ITC towards implementing the concept of smart specialization. The smart city concept is based on research-innovation topics which support the smart specialization focus assumed by Romania, and includes 8 specializations of sub-fields that are interrelated, and covering wider concerns in the R&D component of the cluster: Smart Economy, Smart Technology, Smart Infrastructure, Smart Work and Living, Smart Mobility, Smart Citizen, Cleaner Environment, and Smart Health Care.

### 3.2.2. Research methodology

Social Network Analysis is a sociological paradigm for analysing structural models of social relationships [55–57]. This provides a set of methods and measures for identification, visualization, and analysis of informal personal networks within and between organizations. Thus, Social Network Analysis provides a systematic method of identifying, reviewing, and supporting the exchange of knowledge, ideas, and information on social networks.

From the perspective of knowledge management, SNA helps to position and relate the entire network of the collaborative structure: determine individual expertise, knowledge transfer, and development of better communications [58].

In this context, indexes that are measured within the SNA will be able to determine each of these components. Some authors [59,60] state that the more dense a network, the more innovations will spread faster and easier; the better the network is connected, the faster the dissemination of information; the dissemination of information/knowledge/innovations is done faster in subgroups and slower in groups when there is inter-connectivity; networks that have a component conduct better dissemination of information than networks which have bridges; the speed of adoption of the innovation depends on the power of a node in the network, respectively the node with the highest power will adopt the earliest and fastest innovation.

Starting with the key parameters, defining a smart city concept [22,55,61] the thematic fields of research became in our study the basis for establishing the Groups of Competence (Figure 2).

By addressing these research topics, the Groups of Competence formed out of the teams of researchers for academic research networks will be an essential component in the structure of the Cluj ITC cluster. To achieve this objective, the integration of research themes in the concept of "Smart Cities" will be entirely conditioned on how Groups of Competence cooperate and adopt a common strategy that will lead to the creation of the central research unit (forward named the Centre of Knowledge Transfer for Smart Cities—CKTSC). One of the preconditions was to have at least one theme that includes all groups, and for that the central unit is created, appealing to creativity and maximum opening (Figure 2).

Each Research Structure, which has its own members, will participate in the creation of the centre by approaching research topics that target the components of the smart city. Selection between them of specific topics, oriented towards intelligent specialization, will constitute the resources and the basis of the formation of the research groups, called Groups of Competence. Together they form the unit of competence, called the Centre of Knowledge Transfer for Smart Cities (CKTSC). Their common strategy is a permanent interaction (networking) and finding interdisciplinary solutions that lead to

differentiation through excellence and quality of the innovative solutions, proposed by the ITC Cluster, for improving the sustainability of smart cities.

For this purpose, the Groups of Competences within CKTSC form a social network, where nodes are Groups of Competence and the connections are information, knowledge, and ideas (Table 2).

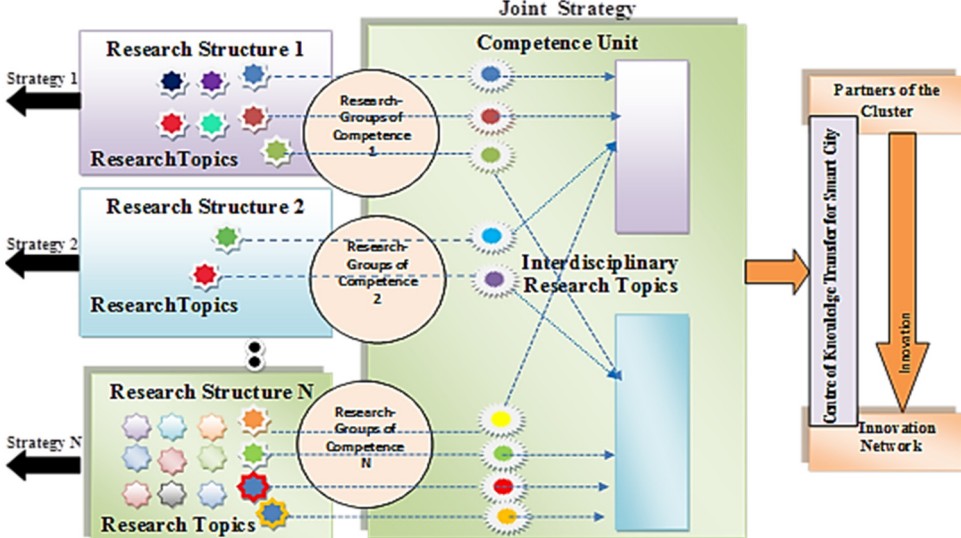

**Figure 2.** The project "Centre of Knowledge Transfer for Smart City—CKTSC". Source: the authors.

**Table 2.** List of actors (nodes) in an innovative cluster of a smart city model. Source: the authors.

| Specific Research Domain/Knowledge Transfer Topics | Groups of Competence | Encoded Subgroups | Corresponding Nodes |
|---|---|---|---|
| Use of intelligent solutions for building/building/housing management such as 4G technology, smart grid systems, establishing high-speed broadband connections | Smart Technologies (ST) | ST-1 | 20 |
| | | ST-2 | 19 |
| | | ST-3 | 21 |
| | | ST-4 | 17 |
| Cultural Spaces and Smart Systems for Formal and Informal Education | Smart Citizens (SC) | SC-1 | 7 |
| Systems of management of transport, energy, water and telecommunications networks | Smart Infrastructure (SI) | SI-1 | 18 |
| | | SI-2 | 16 |
| | | SI-3 | 14 |
| Use of adapted, efficient and secure multi-modal transport systems, including efficient traffic management, electrical infrastructure management, including an Electric Mobility Management System and reducing environmental management costs | Smart Mobility (SM) | SM-1 | 3 |
| | | SM-2 | 5 |
| | | SM-3 | 5 |
| Innovative economic clusters and meso-economic organizational forms Entrepreneurship Networks, Entrepreneurship Promotion Developing strategic alliances among SMEs SME development in economic sectors Economic complexities Developing creative industries Developing social business incubators Developing communities as destinations | Smart Economy (SE) | SE-1 | 9 |
| | | SE-2 | 8 |
| | | SE-3 | 6 |
| Green Smart Buildings, energy efficiency, intelligent buildings with advanced automated infrastructure | Smart Work and Living (SWL) | SWL-1 | 15 |
| | | SWL-2 | 13 |
| | | SWL3 | 12 |
| Urban Ecological Spaces, Brownfields Management | Cleaner Environment (CE) | CE-1 | 10 |
| | | CE-2 | 11 |
| The adoption of efficient and intelligent solutions for managing the health system, from an energy point of view, of the infrastructures of healthcare, the security integrated in the healthcare units, of the connected health services at home | Smart Healthcare (SH) | SH-1 | 1 |
| | | SH-2 | 2 |

In order to conduct the analysis of the network, we completed the following stages:

a.  *Collecting data regarding specific needs and problems.*

At this stage we have selected the research topics related to Smart Cities and establishing the relationship and the competence offered by the Groups of Competences from the Cluj ITC Cluster.

b.　*Outlining and clarifying objectives and the scope of analysis.*

We've aimed in the SNA analysis to determine the topology of the social network of the Cluster's academic members and thus to find the most favourable positions and relationships of the actors within this network by using the designed structure (Groups of Competences); the expected results were to induce and disseminate proper knowledge and innovation flows, so as to enhance the performances of collaborative networking, which, in our opinion, will generate innovative solutions for the development of the smart city. The specific objectives of the analysis were:

✓　to identify stakeholders who have leading roles (launch new ideas, are knowledge brokers, or lead the information);

✓　to identify opportunities for maximizing knowledge flows;

✓　to look for areas with the maximum impact of knowledge exchange.

c.　*Developing a methodology for selecting the data and information that will be the basis of the quantitative analysis of the research and network design. The preliminary steps towards the development of the network were*

✓　establishing nodes (Groups of Competences) and collecting information about relationships within a group of 66 researchers/academics, individual members of the cluster; the selection of the researchers was made according to the research activity and the results related to the smart cities problem during the analysed period;

✓　establishing the Groups of Competences, who are groups of researchers and/or departments of the university members of the cluster, who covered topics related to the concept of the smart city during the period 2007-2017(this period was chosen because it corresponds to the planning period of the Research and Development Strategy of Romania [62]; the evaluation document of the research activity performed by the researchers was called "Integrated File of the Competence Group";

✓　the interrogation of the research database was made for the identification of the common researches of the groups of competence with the purpose of determining the common authors and the number of common scientific products realized during the analysed period; the database allowed us to identify groups of competence according to affiliation and the scientific activities related to the thematic areas of the smart cities project;

✓　these preliminary data were the basis for the continuation of the research, respectively of the analysis itself, which is the subject of this paper. Thus, we have performed the following steps:

✓　We have selected 21 Groups of competences, encoded as in Table 2, according to the criteria described. The number of subgroups and members within the target groups of competence vary; the codification for these subgroups are presented in Table 2, column 3.

✓　In order to identify the relationships and knowledge flows that will form the links of the social network, we determined a relational matrix, based on the integration of the following data: common articles, research papers, books, joint research projects/reports, patents related to the smart cities research topics.

✓　We collected the analytical data and synthesized them by grouping by category of research products the number of common products obtained in each subgroup.

✓　We encoded the subgroups that make up the research themes of the same group. Each research team in the subgroup represents a node with a given number. Their correspondence is presented in Table 2.

The nodes (vertices) are the Groups of Competences; two departments or research partners are in a relationship if they belong to the same Group of Competence. This had been determined after evaluating the "Integrated File of the Competence Group". In this way, a network is undirected and the ties are symmetrical. After establishing members and research teams belonging to a Group of Competence, migration to another group was banned. Therefore, there are no situations where members, departments, or research groups are part of two or more Groups of Competences. That is why the network is considered as weighted, only until the interconnection of the actors was set; it has been a network with directed and dichotomous, unweighted ties. Directed links (ties) are useful for analysing the flows that occur between nodes, such as information/knowledge/innovative ideas flows, etc.

*d.    Mapping out the network visually using a software tool—SocNet V [63] designed for this purpose.*

This part is the body of the analysis which is presented in the following sections.

*3.3. Analysis framework*

To complete "the picture" of the network's parameters studied, we have selected the proper parameters (indexes) of the collaborative network, resulting in the following topology (Figure 3):

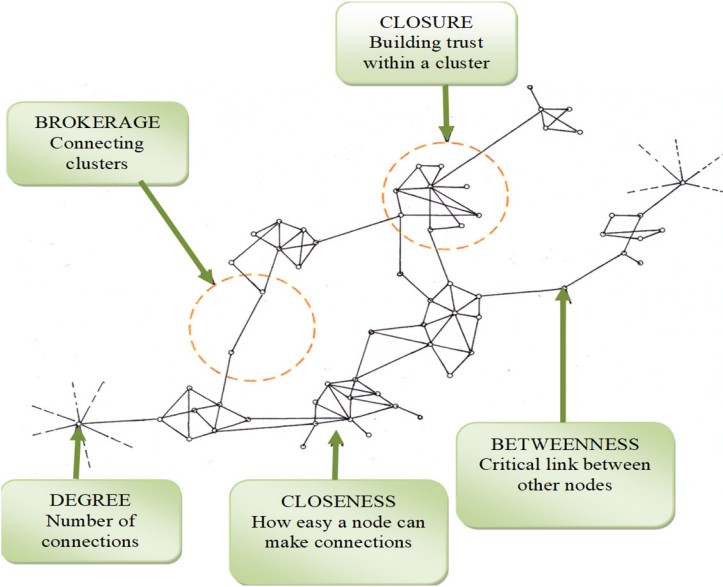

**Figure 3.** The social network. Source: the authors, adapted from [64].

The network is seen as a channel for information dissemination or exercise of influence [60]. Therefore, the social role of the actors depends on both the group to which it belongs and its position within these groups. The "distance" has been developed to quantify the relative positions of the individuals in a network compared to others and to explain the individual performances of the actors that can be quantified by the absolute differences of these values.

## 4. Results

*4.1. Results of the analysis*

The network analytic techniques used were based on the methods indicated by Wasserman and Faust [65]. Data has been processed and analysed through SocNet V software [63].

### 4.1.1. Clustering Coefficient

Using the Clustering Coefficient, we can see how well the neighbours in a graph are connected, while creating the so-called "small-world networks", disconnected from each other; if the value of this coefficient is small, then there are more links between the cliques.

Each Competence Group in Node 7, representing a Group of "Smart citizens" (SC1), has the lowest value of the coefficient and suggests that there are more internal links. This means that there are segments of collaboration between a small number of actors, small networks that overlap those already existing, which actually enrich—on deeper analysis—the network, creating the macro and micro procedural relations between the social actors interconnected in the group of competence space.

The indicator value for the entire network is 0.585, and it reveals that the groups ST, SC, SI, SM, SE, SW, CE, and SH are interconnected, improving the flow of exchanges; thus, the degree of dependence between companies, not individuals, is increased (Table 3).

**Table 3.** Clustering Coefficient. Source: data processing through SocNet V.

| Node | CLC | Node | CLC | Comments |
|------|-----|------|-----|----------|
| 1 | 0.515 | **12** | 0.500 | |
| 2 | 0.457 | **13** | 0.722 | $N_i$, of a node i. In a directed network, the clustering coefficient |
| 3 | 0.716 | **14** | 0.577 | is computed with the formula (Wasserman and Faust, 1994): |
| 4 | 0.579 | **15** | 0.721 | $C_i = \frac{|\{e_{jk}:v_j,\ v_k \in N_i,\ e_{jk} \in E\}|}{k_i(k_i-1)}$ |
| 5 | 0.647 | **16** | 0.614 | In undirected networks, the formula is: |
| 6 | 0.509 | **17** | 0.527 | $C_i = \frac{2|\{e_{jk}:v_j,\ v_k \in N_i,\ e_{jk} \in E\}|}{k_i(k_i-1)}$ |
| 7 | **0.421** | **18** | 0.518 | An edge $e_{jk}$ connects vertex $v_j$, with vertex $v_k$, $k_i$ the number of vertices, |
| 8 | 0.605 | **19** | **0.829** | $N_I$, in the neighbourhood, $N_I$, of a node i, E set of vertices |
| 9 | 0.651 | **20** | 0.568 | **Node 19 has the maximum Clustering Coefficient: 0.829** **Node 7 has the minimum Clustering Coefficient: 0.421** |
| 10 | 0.599 | **21** | 0.480 | ACC = Average Clustering Coefficient |
| 11 | 0.538 | **ACC** | **0.585** | |

### 4.1.2. Centrality

In order to observe a progressive evolution of innovation networks, which means a process of network reconfiguration that is necessary to evaluate the location of each actor in the network. Measuring the location means finding the degree of centrality of a node (i.e., its importance and prominence).

Network centrality is a very important parameter for understanding the power and hierarchy of competence groups. Power within a network depends on relationships. This means that social actors, who have positions with more opportunities, alternative ways to access information, and fewer constraints, are stronger than others and have higher network centrality. The actors of a network that are closer to most other actors are considered as models and also possess a remarkable power in the network. The actors in the centre of the network are stronger than those in the marginal areas; those who play the role of intermediaries or bridges between other actors also have considerable power within the network.

The relationship between the centrality of the nodes can tell us a lot about the overall structure of a network. Node 15—Smart Working and Living (Table 4), sub-group 1—is the strongest group in the entire network. We found that the most important project ideas have emerged from this Group of Competence. The hypothesis of the study is thus verified: The greater the power of a node in the network, the sooner it will adopt an innovation. Results are presented in Table 4, Diagram 4.

**Table 4.** Power Centrality (PC) **. Source: data processing through SocNet V.

| Node | PC | PC′ | %PC′ | Node | PC | PC′ | %PC′ | Comments |
|------|-----|------|------|------|------|-------|------|----------|
| **1** | 13 | 0.65 | 65 | **12** | 19.67 | 0.483 | 48.3 | |
| **2** | 10.2 | 0.508 | 50.8 | **13** | 13 | 0.65 | 65 | The PC index of a node u is the sum of |
| **3** | 17.5 | 0.875 | 87.5 | **14** | 12 | 0.6 | 60 | the sizes of all Nth-order |
| **4** | 16 | 0.8 | 80 | **15** | 20 | 1 | 100 | neighbourhoods with weight 1/n. PC′ is the standardized index: The PC |
| **5** | 17.5 | 0.875 | 87.5 | **16** | 11.8 | 0.592 | 59.2 | score divided by the total number of |
| **6** | 9 | 0.45 | 45 | **17** | 11.7 | 0.583 | 58.3 | nodes in the same component minus 1 |
| **7** | 14 | 0.7 | 70 | **18** | 18.5 | 0.925 | 92.5 | (SocNet V Manual). PC range: 0 < PC < 20 (star node) |
| **8** | 14 | 0.7 | 70 | **19** | 15.5 | 0.775 | 77.5 | PC′ range: 0 < PC′ < 1 |
| **9** | 16.5 | 0.825 | 82.5 | **20** | 16 | 0.8 | 80 | Max PC′ = 1 (node 15) |
| **10** | 17 | 0.85 | 85 | **21** | 15.5 | 0.775 | 77.5 | Min PC′ = 0.45 (node 6) |
| **11** | 10.2 | 0.508 | 50.8 | - | - | - | - | |

**Diagram of Power Centrality**

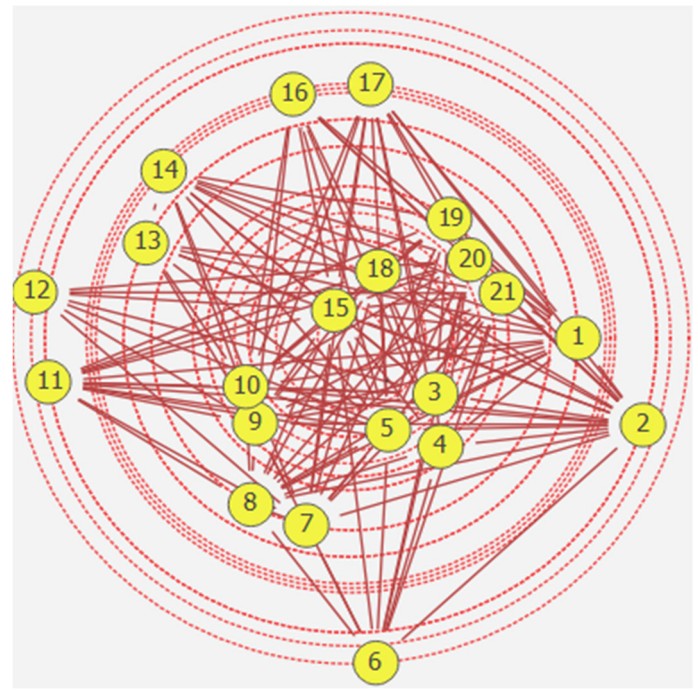

** The Power Centrality (PC) is a generalised degree centrality measure suggested by Gil and Schmidt. For each node u, this index sums its degree (with weight 1), with the size of the 2nd-order neighbourhood (with weight 2), and in general, with the size of the k-ordered neighbourhood (with weight k). SocNet V Manual.

In this case we have a network with a high degree of centralization, in which very few nodes with a central position dominate (Nodes 15, 17, 19, 21). The network will fragment very quickly into small, unconnected groups, if the nodes are destroyed/damaged or eliminated. However, the nodes located right at the centre can be points of weakness, as the high degree of centralization has prolific effects on projects which need highly skilled human resources (Figure 3).

### 4.1.3. Degree Centrality

According to Miler-Prothmann [58], the degree centrality is an indicator that measures through the input and output links is held by a member of the network, the power and expertise of that actor. If the data are not symmetrical then the parameter measures the popularity of that actor (member)

within the network; those who have high levels of expertise, have also many out-degree connections and are considered to be particularly influential in the network. The Group of Competence Smart Work and Living, subgroup SWL-3, has again the maximum degree of centrality. This group is the driver of knowledge in the proposed "Smart Cities" project (Table 5, Diagram 5).

**Table 5.** Degree Centrality ***. Source: data processing through SocNet V.

| Node | DC | DC' | %DC' | Node | DC | DC' | %DC' | Comments |
|------|----|-----|------|------|----|-----|------|----------|
| **1.** | 6 | 0.3 | 30 | **12.** | 2 | 0.1 | 10 | |
| **2.** | 3 | 0.15 | 15 | **13.** | 6 | 0.3 | 30 | |
| **3.** | 15 | 0.75 | 75 | **14.** | 4 | 0.2 | 20 | To compute or Group Degree Centrality, SocNet V uses the Freeman's formula for unvalued graphs (SocNet V Manual). |
| **4.** | 12 | 0.6 | 60 | **15.** | 20 | 1 | 100 | |
| **5.** | 15 | 0.75 | 75 | **16.** | 4 | 0.2 | 20 | $GDC = \frac{\sum(maxDC'-DC')}{(N-1)x(N-2)/(2xN-1)}$ |
| **6.** | 1 | 0.05 | 5 | **17.** | 5 | 0.25 | 25 | DC' is the standardized. DC DC range: $0 < C < 20$ |
| **7.** | 8 | 0.4 | 40 | **18.** | 17 | 0.85 | 85 | DC' range: $0 < C' < 1$ |
| **8.** | 8 | 0.4 | 40 | **19.** | 11 | 0.55 | 55 | Max DC' = 1 (node 15) |
| **9.** | 13 | 0.65 | 65 | **20.** | 12 | 0.6 | 60 | Min DC' = 0.05 (node 6) |
| **10.** | 14 | 0.7 | 70 | **21.** | 11 | 0.55 | 55 | |
| **11.** | 3 | 0.15 | 15 | - | - | - | - | |

**Diagram of Degree Centrality**

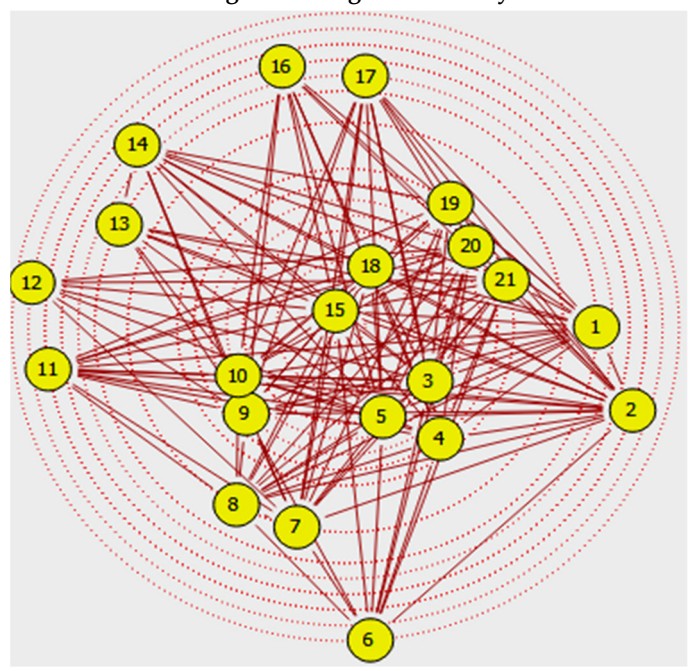

*** In undirected graphs, the DC index is the sum of edges attached to a node u. In digraphs, the index is the sum of the outbound arcs from node u to all adjacent nodes. If the network is weighted, the DC score is the sum of weights of outbound edges from node u to all adjacent nodes. SocNet V Manual.

### 4.1.4. Closeness centrality

Closeness is the theoretical distance of a given node to all other nodes. This is frequently used in the study of knowledge diffusion and innovation; this parameter takes into account the indirect connections, as opposed to the Degree centrality. In the directed graphs, the output arcs of the network are related to the number of walks an actor must reach to the other actors (Table 6, Diagram 6). In flows,

this measure is interpreted as a time index until a new flow arrives within the network (SocNetV Manual, 63)

**Table 6.** Closeness Centrality (CC) *****. Source: data processing through SocNetV.

| Node | CC | CC′ | %CC′ | Node | CC | CC′ | %CC′ | Comments |
|------|------|------|------|------|------|------|------|----------|
| 1. | 0.0294 | 0.588 | 58.8 | 12. | 0.05 | 1 | 100 | |
| 2. | 0.0222 | 0.444 | 44.4 | 13. | 0.0294 | 0.588 | 58.8 | The CC index considers outbound arcs only and isolate nodes are dropped by default. Group CC is calculated using Freeman's general formula, in undirected graphs: $GCC = \frac{\sum(maxCC'-CC')}{(N-1)x(N-2)/(2xN-1)}$ CC range: 0 < C < 0.05 CC′ range: 0 < C′ < 1 Max CC′ = 1 (node 12) Min CC′ = 0.417 (node 6) CC range: 0 < C < 0.05 |
| 3. | 0.04 | 0.8 | 80 | 14. | 0.0278 | 0.556 | 55.6 | |
| 4. | 0.0357 | 0.714 | 71.4 | 15. | 0.0217 | 0.435 | 43.5 | |
| 5. | 0.04 | 0.8 | 80 | 16. | 0.027 | 0.541 | 54.1 | |
| 6. | 0.0208 | 0.417 | 41.7 | 17. | 0.025 | 0.5 | 50 | |
| 7. | 0.0312 | 0.625 | 62.5 | 18. | 0.0435 | 0.87 | 87 | |
| 8. | 0.0312 | 0.625 | 62.5 | 19. | 0.0345 | 0.69 | 69 | |
| 9. | 0.037 | 0.741 | 74.1 | 20. | 0.0357 | 0.714 | 71.4 | |
| 10. | 0.0385 | 0.769 | 76.9 | 21. | 0.0345 | 0.69 | 69 | |
| 11. | 0.0222 | 0.444 | 44.4 | - | - | - | - | |

**Diagram of Closeness Centrality**

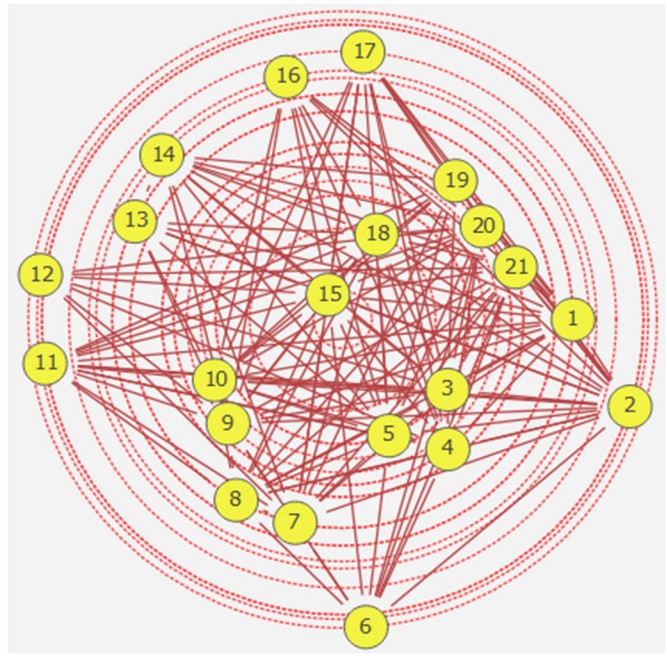

***** The CC index is the inverted sum of geodesic distances from each node u to all other nodes. CC′ is the standardized CC (multiplied by N-1 minus isolates). SocNet V Manual.

The information, viewed as an interpretation of this parameter, will follow the shortest path on which all the paths, including the shortest ones, have been crossed simultaneously. This measure works only in the strongly connected networks (Diagram 5). Within the cluster, it is represented by researchers who cope with intelligent buildings in the group Smart Work and Living, sub-group SWL-3, node 12.

### 4.1.5. Betweenness Centrality

Betweenness centrality is a parameter that shows the limits to which the position of a network actor falls on the geodesic paths between the other actors, thus being a measure by which knowledge

brokers or gatekeepers of that network can be identified [63–67]. The Smart Infrastructure group (SI-1) is the broker (maximum BC' = 0.234) between two groups of the network and thus plays a strong role within it (Table 7, Diagram 7).

**Table 7.** Betweenness Centrality (BC) ******. Source: data processing through SocNet V.

| Node | BC | BC' | %BC' | Node | BC | BC' | %BC' | Comments |
|------|------|--------|-------|------|-------|----------|--------|----------|
| 1. | 13.7 | 0.0362 | 3.62 | 12. | 0.254 | 0.000668 | 0.0668 | |
| 2. | 5.94 | 0.0156 | 1.56 | 13. | 0.893 | 0.00235 | 0.235 | The BC index of a node u is the sum of delta (s,t,u) for all s,t in V where delta (s,t,u) is the ratio of all geodesics between s and t which run through u. |
| 3. | 6.6 | 0.0174 | 1.74 | 14. | 0.589 | 0.00155 | 00.155 | |
| 4. | 13.7 | 0.0361 | 3.61 | 15. | 6.13 | 0.0161 | 1.61 | |
| 5. | 5.08 | 0.0134 | 1.34 | 16. | 0.7 | 0.00184 | 0.184 | BC Range = Number of pairs of nodes excluding u |
| 6. | 0 | 0 | 0 | 17. | 2.53 | 0.00666 | 0.666 | C' is 1 when the node falls on all geodesics |
| 7. | 27.6 | 0.0727 | 7.27 | 18. | 88.9 | 0.234 | 23.4 | BC' is the standardized BC. |
| 8. | 3.97 | 0.0105 | 1.05 | 19. | 0.754 | 0.00198 | 0.198 | BC range: 0 < BC < 380 |
| 9. | 3.95 | 0.0104 | 1.04 | 20. | 7.98 | 0.021 | 2.1 | BC' range: 0 < BC'< 1 |
| 10 | 18.3 | 0.0481 | 4.81 | 21. | 60.1 | 0.158 | 15.8 | Max BC' = 0.234 (node 18) |
| 11. | 1.2 | 0.00315 | 0.315 | - | - | - | - | Min BC' = 0 (node 6) |

**Diagram of Betweenness Centrality.**

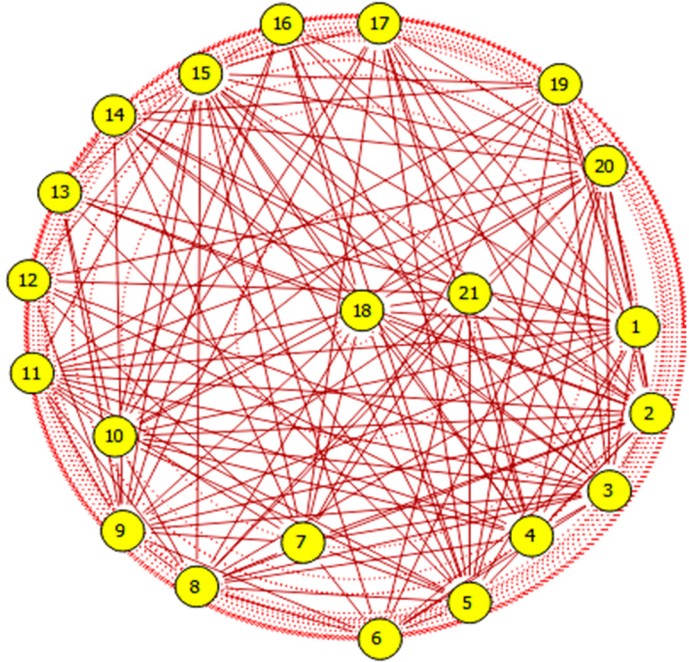

****** The BC index of a node u is the sum of delta (s,t,u) for all s,t in V where delta (s,t,u) is the ratio of all geodesics between s and t which run through u. SocNet V Manual.

SI1 finds the shortest paths and it is closer to others, having a privileged position, and the overall image of the processes that take place in the network is monitored through the flow of information that this actor's favourable position holds.

Synthesizing the results obtained from the SNA analysis, the following discussion is presented:

✓ The connectivity of the competence groups is high, the Clustering Coefficient index across the whole network being 0.585. The greatest potential for clustering is subgroup ST-2 (node 19), Smart Technologies, a group that addresses research topics that can be integrated within the other groups, attracting research, projects, or patents related to the most dynamic solutions

for the development of a smart city, such as Smart Home, high speed broad band connection, 5G Technology, etc.

✓ Power Centrality is highest in the Smart Work and Living (SWL-1) competence subgroup. This group could have maximum influence in the direction of collaboration with the other groups and decision-making power in solving the smart city's research problems; it also has the maximum expertise on sustainable solutions in the smart city, such as "green buildings", "energy efficiency", "smart buildings", or "automated infrastructures".

✓ Degree Centrality has maximum value in subgroup three of the Smart Work and Living competence group (SWL-3). This group is formed from experts in the research areas of sustainability of the smart city. This subgroup could ensure the transfer of knowledge and technologies for sustainability to other entities within or outside the cluster; it could be a trainer as well as a consultant.

✓ Within the subgroup SWL-3, Closeness Centrality is also at a maximum, which suggests that this group plays the role of supporting the collection and processing of information from the external environment of the smart city; the group investigates issues related to GIS (Geographic Information System), urban marketing, spatial development, etc.

✓ Subgroup SI-1—Smart Infrastructure—is the knowledge broker within the cluster, because Betweenness Centrality in this subgroup is at a maximum. This indicates that the group could ensure the rapid dissemination of information and knowledge related to smart city research within the network. The subgroup is also the one that could lead the research platform within the cluster.

✓ Through Social Network Analysis, we have determined the metrics and topology of the Clustering Coefficient, Density, Power Centrality, Degree Centrality, Closeness Centrality, and Betweenness Centrality. The maximum values of these indexes indicate the role of each group in the collaborative smart city network (Table 8):

**Table 8.** Role of the Group of Competences in the collaborative network.

| Group of Competence | SNA Index | Role of the Actor in the Network |
|---|---|---|
| ST-2/Smart Technologies | Clustering Coefficient | Max. clustering potential |
| SWL-1/Smart Work and Living | Power Centrality | Max. influence and decision power |
| SWL-3/Smart Work and Living | Degree Centrality | Max. knowledge driver potential |
| SWL-3/Smart Work and Living | Closeness Centrality | Max. external information flow input |
| SI-1/Smart infrastructure | Betweenness Centrality | Max. diffusion of information potential |

### 4.2. Limitations of Research

Without specific knowledge about the network dynamics and organizational behaviour, its analytical values are essentially un-interpretable in relation to different periods of analysis or used for the purpose of comparison of similar situations. The predictions of organizational changes are low. The practical view of the networks used as a source of information in decision making can be successfully combined with more traditional research which, in turn, can be used recursively to interpret the smart city collaborative network model.

### 4.3. Contributions to the Research

The smart cities concept exploited in our research, based on multi-stakeholder collaboration and networking with emphasis on interconnections of different aspects of urban life, is proving the importance of having a holistic view on sustainability for smart cities (in a mixed format), although it is often debated in the literature of the last years and new challenges regarding these approaches are still arising. The contribution of our research to the completion of knowledge in this field has both theoretical and managerial implications.

### 4.3.1. Theoretical implications

Our research identifies and formulates in a novel and original way that a concept of a sustainable city as a smart organization can be considered as a complex and adaptive system, which is functioning on the base of permanent adaptations of human actions towards a desired sustainable state, resulting from a city's behaviour as a whole system and consisting of natural, economic, social, and other involved components.

The adaptability here is interpreted as dynamic learning and organizational processes in networks of Groups of Competences, which through permanent structural improvements accumulate the necessary knowledge to serve or/and directly solve the arising issues of smart sustainable cities. The Groups of Competences in this way generate the additional economic, social, and environmental values within the adaptive smart sustainable city model and enhance its smart entrepreneurship. Thus, the suggested new paradigm confers our research added value to holistic philosophy and to organizational and learning designs of a smart sustainable city.

The holistic philosophy of a smart sustainable city was shaped through using the Complex Adaptive System theory (CAS) as a dynamic frame for understanding a network of agents that collaboratively could form a smart city model, and thus respond to sustainability goals. The sustainable move will depend on how much the different components of a smart city are driven by, can adapt to, and contribute to common sustainability goals.

The notion of organizational design of a smart sustainable city was developed through suggestion and further interpretations of Groups of Competences as the contouring and centring nodes of collaborative networking, generating its format of a system, complexity, adaptability, and larger resulting power of working together, which reflect the features of the CAS model.

Enhancing the learning concept of a smart city was provided by adaptive heterogenic and interdisciplinary networking of the Groups of Competence connected to sustainability goals. Their members are from different fields, grouped by their competences in the main dimensions of development of the smart sustainable city. They work together in high interconnectivity, having common goals towards excellence in innovative and smart solutions through the diffusion of knowledge.

The model was developed and technically tested by the means of SNA on the base of an innovation cluster and resulted in defined Groups of Competences aimed at addressing innovative and sustainable solutions through applied R&D, which underpin "Smart" components of cities (Smart Economy, Smart Technology, Smart Infrastructure, Smart Work and Living, Smart Mobility, Smart Citizen, Cleaner Environment, and Smart Health Care).

### 4.3.2. Managerial implications

Participating in collaborative innovative networks within the smart city frame, serve both the community sustainability goals and, through acceptable innovations, better survival in the market—if this is a business type of partner. Thus, on a higher level, we see the benefits of diversity and complementarity of the network together with the intense connections between the activities involved. Thereby, the results are of a real help to all those involved in solving the practical problems of smart cities: local authorities, business managers, public institutions, and non-governmental organizations that are considering innovation in smart cities through the Smart Cities Collaborative Network; the conceptual model allows its replication under different conditions, by testing, validating, and co-creating innovations in smart cities. Understanding the smart city collaboration networks and the relationships within its behaviour can help stakeholders to decide what they need and, thus, stimulate their business and innovation goals with the help of the Smart Cities Collaborative Network. These collaborative forms will have the effect of intensifying, diversifying, and increasing the quality of innovation in the smart city.

## 5. Conclusions

The sustainability concept within a smart city refers to the process of balancing different sectors (traditionally, economy, society, and environment) by creating wide possibilities of cooperation between different public and private actors, from the academic environment and from the large community, which can work together to ensure the efficient integration of multiple and/or complementary perspectives on sustainable urban intelligent development [5,7,11]. Studying the complex adaptive system for designing the concept of a smart sustainable city, represents a special challenge for researchers, with major practical implications for the efficiency of governments and development of policies of cities and regions around the world. Deployment of such an approach in an organized and formalized framework will lead to efficient integration and sharing of knowledge.

The research aimed to develop a pattern of the collaborative structures as an organizational frame for smart and sustainable city development. Our paper, by answering the research questions formulated, contributes to the completion of the knowledge in this field.

The paper covers both theoretical and field investigations.

The theoretical analysis revealed the possibility of using a concept of complex adaptive systems for building a flexible network of competence groups with a fast reaction on emerging sustainability needs. For this, (1) the selected smart city approaches are generalised, in order to interpret a complexity of a smart city concept; and (2) an interpretation of knowledge networks as dynamic adaptive tools in the concept of a learning city is also provided.

The theoretical part of the research also included the contextualizing of the CAS concept within the smart sustainable city. The collaborative structures within the model are the Groups of Competences, which act as agents of change, transformation, and innovation towards achieving the goals of sustainability along with smart city activities. The adaptation happens when the parts involved in the system react on internal and external changes, pushing a system as a whole to become self-organized in the new quality of dynamic networking, to together solve an appearing problem or to use the emerging possibilities. Thus, the model was designed in a way that integrates and customizes that area of the system through which its supportive structures (Groups of Competences) ensure substantial collaborative networking, which will generate innovative solutions for the development of the smart city.

The Groups of Competence are centred at the universities' research departments which are the members of the innovative cluster; their networking facilitates the innovation processes in the fields that form the thematic picture of the smart city collaborative model. Using both internal and external networks, knowledge management in this model becomes more effective.

The field part of the research went down to the meso- and microeconomic levels, in which the collaborative structures (Groups of Competences) were analysed by the SNA method. The analysis carried out in this paper is based on a study relating to an innovative cluster in Romania; the purpose was to find the most favourable positions and relationships of the actors within this network. As a result of the Social Network Analysis, the experts who can act as the central brokers or multipliers of useful information and knowledge were identified. This managerial component of the research allows us to determine the specific indexes within the SNA, which help to identify the most influential groups and subgroups, their dominant positions, their powers and their degrees of cohesion, and to obtain the graphical visualization of each calculated index.

The results of research serve as a basis for the continuous improvement of the managerial process, within an innovative cluster. The Social Networks Analysis can also contribute to decisions on defining the key competencies and the professional development of human resources in the innovation network that is taken as a base for the smart city collaborative model. In terms of inter-organizational processes, particularly building the innovation network, the Social Networks Analysis provides a powerful tool. Social Networks Analysis is based on rapid and effective identification of interventions that can be instrumental in facilitating communication processes and community activities to enhance a covering

scale of knowledge exchange and to improve informal inter-organizational relationships for better knowledge-sharing in an innovative cluster.

Comparing with previous research, we can summarize several distinctive features of and results obtained in our work: (1) the principles of the Complex Adaptive System served as a frame for defining the logic of knowledge groups' adaptive dynamic within a smart sustainable city; (2) Social Network Analysis was used to define the Groups of Competence and their networking; (3) articulating both the concepts of CAS and the metrics of SNA, it was possible to make the conclusions on the higher power of networked/organized Groups of Competence than their simple sum; and (4) the interpretations of possible managerial applications of SNA results were provided.

In future research, it will be useful to extend the organizational modelling of the Competence Groups' adaptive networking for further enhancing the balance of sustainable goals within a smart city concept. The success of the smart city collaborative model will depend on how the social structure of the network (CKTSC) is shaped and modified to fit the thematic sustainable objectives of the community.

**Author Contributions:** C.M.R. and S.S. conceived and designed the conceptualization; A.T.R., G.D.B. and D.-C.T. performed the research and the analysis; R.T. completed the investigation and methodology. All authors contributed in discussing the research, writing parts of the paper and commenting on draft versions and finalized the paper. All authors have read and agreed to the published version of the manuscript.

**Funding:** This research received no external funding.

**Acknowledgments:** The authors wish to thank Marion Heredia and Marnie Howlett for their helpful comments on English grammar.

**Conflicts of Interest:** The authors declare no conflict of interest.

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
