# Peer review of "A Pattern of Collaborative Networking for Enhancing Sustainability of Smart Cities"

_sustainability, doi:10.3390/su12031042_

Round 1

Reviewer 1 Report

Overall a good paper and I recommend it for publication pending a few requests: please do a grammar check, and enhance the resolution of your figures.

Author Response

Dear Reviewer,

We are thankful for your comments and feedback of our submitted article. We have modified the article significantly following yours and the other reviewers’ suggestions. Below you can find our detailed response to your specific observations, we have used the template suggested by the journal to answer each point of your evaluation. We hope we have addressed all your concerns and observations.

Response to Reviewer 1 Comments

Point 1:Please do a grammar check

Response 1:The grammar check of the paper was done by a native English speaker

Point 2:Enhance the resolution of your figures.

Response 2:

The quality of all the figures and tables was improved(please see in the manuscript).

Figure 1.Collaborative Smart City Model, Figure 2.Strategy of the Project ’’Centre of Knowledge Transfer for Smart City-CKTSC”,Figure 3.Topology of Social Network

Also, at the 4th part of the Results, we chose to incorporate the diagrams of all the calculated parameters, in the related tables, for a more compact and suggestive representation

Table 4. Power Centrality (PC)**Source: data processing through SocNetV, Table 5. Degree Centrality (DC)***, Source: data processing through SocNetV, Table 6.Closeness Centrality (CC)****, Source: data processing through SocNetV Table 7.Betweeness Centrality (BC)*****, Source: data processing through SocNetV.

And a new one: Table.8 Role of the Competences in the Collaborative Network

Reviewer 2 Report

Dear author,

As a reviewer, I left some comments on your file. Please consider and revise the article based on them, especially in regard with references.

Your objectives are not academically written,

Lack of references in proper place,

Conclusion can be rewritten based on new objectives and research questions.

On positive side, your network analysis s conducted correctly.

BEst of luck

Author Response

Dear Reviewer,

We are thankful for your comments and feedback of our submitted article. We have modified the article significantly following yours and the other reviewers’ suggestions. Below you can find our detailed response to your specific observations, we have used the template suggested by the journal to answer each point of your evaluation. We hope we have addressed all your concerns and observations.

Response to Reviewer 2 Comments

Point 1:

A.Introduction:Some of sentences require citation. These citations must be made when referring to research (such as the second sentence).

Response 1:(the paragraph is changed as below, new lines 32-48):

A smart city idea has been evolving over recent years, which is witnessed by a bunch of research, devoted to definition analysis of smart city concepts [1-7].In their works it is highlighted, that today, a smart city concept is considered both as a theoretical approach (a gradual movement from focus on IT modernisation of infrastructure in the late 1980-sto holistic approach of city sustainability in two decades) and as practical precedents in different formats[8], indicating the good implemented cases across the world and many places with smart-city strategies. Digital technologies and networks are among the main fundamentals for smart city development [1, 7, 8]. Increasingly sophisticated and rapid changes in technology and social media have led to unprecedented structural changes, especially in urban areas, because the traditional methods and approaches cannot provide such effective and viable solutions. It therefore requires consideration of intervention mechanisms to support fair and sustainable community progress, using the innovation systems. In the relevant literature particular attention is given to the question of increasing viability through innovation systems [9,10]; but the issues on how to improve the capability to structure the city sustainability format [11], and the competence flow in the smart city model, remain fragmented. As an attempt to filling this gap, the paper suggests developing a smart city concept, based on a collaborative research networking, which is leveraged by emerging sustainability goals, rather than separate work in selected sustainability directions within a smart city frame.

Introduction is improved, additional references are included as below (with reference numbers in the list of literature): 

Hollands, R. G. Will the real smart city please stand up? Intelligent, progressive or entrepreneurial?. City, 2008, 12(3), pp.303-320. Zhao, L.; Tang, Z. Y.; Zou, X. Mapping the Knowledge Domain of Smart-City Research: A Bibliometric and Scientometric Analysis. Sustainability, 2019, 11(23), 6648. Lazaroiu G. C.;Roscia M., Definition methodology for the smart cities model.Energy,2012,47(1);pp. 326–332. Albino, V; Berardi, U.; Dangelico, M. A., Smart cities: Smart Cities: Definitions, Dimensions, Performance, and Initiatives.Journal of Urban Technology, 2015, 22(1); pp.3–21. Available online at:https://www.researchgate.net/publication/275042309_Smart_Cities_Definitions_Dimensions_ Performance_and_Initiatives, ( accessed on 22 October 2018). Monzon, “Smart Cities Concept and Challenges: Bases for the Assessment of Smart City Projects,” in M. Helfert et al. (Eds.): Smart greens 2015 and Vehits 2015, CCIS 579, pp. 17–31, 2015. DOI: 10.1007/978-3-319-27753-0_2 Visvizi, A.; Lytras, M. D. Smart Cities: Issues and Challenges: Mapping Political, Social and Economic Risks and Threats,Elsevier,2019.

B.Also there are some claims in the preface which require a reference (such as lines 46 to 52) and the author cannot address them personally.

Response :

(new lines 49-56)

The development of the smart city is becoming an increasingly multiple challenge that has appeared in the urban environment. Rapid changes of priorities and perspectives of cities, determines an integrated approach of information, knowledge, technologies, processes that have to be articulated in a systemic vision. In this context, urban planners are launching more and more projects based on technologies, to respond to challenges of city expansion and finding the proper infrastructural solutions. This means that technology, research and innovation should no longer be a target, but become a core component of the smart city, forming its adaptability to emergent changes[1,4,12].

Point 2:Main objective is normally just one, and some related questions. Here are four objectives.

Then, an objective must be measurable. here it is not.

for example,  the first goal is not an objective. smart city system has been interpreted several times as a complex system.

Response 2:

The research questions are reshaped (lines 85 – 92); the gap in research on collaborative concept of smart city is highlighted (lines 72-76).

(New line 77-80):

Thus, the aim of the paper is to substantiate a collaborative pattern of knowledge networking with focusing on sustainability goals within a smart city concept, using logic of a complex adaptive system (CAS). In this way, the internal knowledge strengths are used for moving towards community sustainability.

(New line 85-92):

For achieving this goal, the study was focused on the following research questions:

What theoretical basics fit sufficiently for defining a smart-city frame, capable of solving complexity and adaptability issues of sustainability? b) How to configure a collaborative networking model, which integrates the CAS characteristics of the smart city? c) How can the connections and relationships of a network be established empirically, so as to generate a flow of knowledge, ideas and solutions of excellence for a smart city? d) What are the benefits of configuring Groups of Competences within an innovative cluster, by means of SNA?

Point 3:

Section 2.The literature on this area is weak and needs to be strengthened.

Most of the sentences in this section do not include reference, whereas literature review of the subject is not possible without reference. Add the necessary references.

Response 3:

We’ve completely reorganized the second section of the paper,by combining some secondary paragraphs, in order to be able to develop the main topic related to the CAS, in a comprehensive, coherent way, and having a proper scientific outfit.

For this reason, we considered that the paragraphs that were presented a previous version should be integrated into the newly created paragraphs and, re-organized within this section, without diminishing the importance of the main topic, related to the case of smart cities.

Also in this section, we’ve rewritten the parts that directly quoted the notions addressed, and we've rephrased them, at the suggestion of the reviewers, at the same time introducing new important bibliographic references.

In this way, we obtained a section that bases the theoretical aspects of the research through a unitary, compact and logical character.

Point 4: Image is in low-quality and must be improved.

Response 4:

The quality of all the figures and tables was improved(please see in the manuscript).Also, at the 4th part of the Results, we chose to incorporate the diagrams of all the calculated parameters, in the related tables, for a more compact and suggestive representation.

Point 5:In conclusion the findings should be compared with those of previous research. The conclusion should be amended accordingly.

Response 5:

New line 6119-620

The research results are compared with those of previous research. The future possible directions of research are indicated.

“Comparing with previous research, we can summarize several distinctive features of and results obtained in our work: 1) the principles of Complex Adaptive System served as a frame for defining the logic of knowledge groups’ adaptivedynamic within a smart sustainable city; 2) Social Network Analysis was used to define the Groups of Competence and their networking; 3) articulating both the concepts of CAS and the metrics of SNA, it was possible to make the conclusions on the higher power of networked/ organized Groups of Competence than their simple sum; 4) the interpretations of possible managerial applications of SNA results were provided. 

In future research, it will be useful to extend an organizational modeling of the Competence Groups’ adaptive networking for further enhancing the balance of sustainable goals within a smart city concept”.

Reviewer 3 Report

The paper discusses on an important topic, smart city, from a high level perspective.  It is good to see the discussion and presentation in an abstract and systematic way, but the tech and scientific parts are quite limited of this work. 

The presentation is quite lengthy.  A significant portion of the paper is by directly quoting the sentences in the existing works and reports.  I doubt if this is appropriate; at least the language style (by quoting sentences/paragraphs directly) varies frequently and leads to much distraction.   

For the minor parts, the paper requires much editing.  For example, "considered a." in the abstract; overlap between Line 84 to 102; "expertise,have"; wrong usage of " in page 2.  The authors are suggested to go through the manuscript more carefully.  

Author Response

Dear Reviewer,

We are thankful for your comments and feedback of our submitted article. We have modified the article significantly following yours and the other reviewers’ suggestions. Below you can find our detailed response to your specific observations, we have used the template suggested by the journal to answer each point of your evaluation. We hope we have addressed all your concerns and observations.

Response to Reviewer 3 Comments

Point 1:

A.The tech and scientific parts are quite limited of this work. 

Response 1:

The improvement of the Chapter 3,4 and 5 was made taking into consideration that the empirical, applied research, which we have carried out within a cluster, being specific to managerial research, aimed  to offer solutions for improving the competitiveness of the cluster ,by increasing the performance of the management of information systems within it. In respect with this aspects ,we’ve  reshaped the methodology and research methods .

To improve and clarify this section, we restructured it and completed it, by introducing new paragraphs that present in a clear and logical sequence the stages of this managerial research, as follows (lines 238-399):

1-we made an introduction that delimited the research approach,

2 - introduced a subchapter in which we presented:

The object of the research (Cluj ITC); Description of the quantitative and qualitative research stages. In this stage we’ve introduced all the preliminary steps through which we designed the elements of the present study that aim at analyzing the social networks within the cluster; we modified, for a better understanding, the presentation of the stages that led to the network design;

3-compared to the previous version we introduced the explanation of the parameters calculated for each group (node),

4- introduced the calculation relations of the parameters, the explanation of the terms of the calculation relation, the interval of their framing ,maximum and minimum  limites  of the measured indexes.

5- we’ve  completely modified the graphic quality of all the figures and tables, and at the 4th part ‚’’Results’’, we chose to incorporate the diagrams of all the calculated parameters, in the related tables, for a more compact and suggestive representation.

6- In the results, we included discussions and comments of the qualitative analysis, by which the result of each index is associated with the managerial effects, that  implies the increasing  of the performances of the cluster.

7-Also, the modification of the conclusions was made in accordance with the answers to the research questions addressed in the introduction and, the results obtained from the analysis, emphasizing the importance and the positive practical effects of the study.

Changes are marked with blue.

Point 2:The presentation is quite lengthy.A significant portion of the paper is by directly quoting the sentences in the existing works and reports.

Response 2:

a.The presentation has been shortened.

As for sections 1 and 2, we have modified them according to your requirements.Thus, we completely reorganized the second section of the paper, by combining some secondary paragraphs, by which a shortening of the presentation was made. For this reason, we considered that paragraphs that were presented in the previous version, should be integrated in the newly created paragraphs and re-organized within this section, without undermining the importance of the main topic, related to the case of smart cities.

b.Also in this section, we’ve rewrote the parts that directly quoted the notions addressed and we re-phrased them, at the suggestion of other reviewers too,and, at the same time we’ve introduced new important bibliographic references. Thus,we think ,that the research gains  a unitary, compact and logical character.

Point 3:Editing issues

Response 3:

The editing  and grammar check were made by a native English speaker.

Round 2

Reviewer 2 Report

Dear authors,

Thank you for your revision.

The quality of this article is significantly improved. the methodology is described in a decent way and the findings are presented in acceptable level. more important, fair use of citations cover the problem of unproved claims.

I have no more comments and I cross my fingers for your article.

Best of luck!

Author Response

Dear Reviewer,

 We would like to express our thanks for appreciations of our research results.

We are thankful for all your previous comments and feedback of our submitted article.

Thank you for your time devoted to our paper and for interesting learning experience!    

Reviewer 3 Report

The manuscript has been improved.  The fundamental problems like novelty and scientific value are still there, although the problems have been addressed to some extend in the revised version.  The rest parts are fine.

Author Response

Dear Reviewer,

We are thankful for your second comments and feedback of our submitted article.

We want to express our sincere and respectfull thank-you for your involvement into making our paper  improved. Follwing your suggestions we exposed more details of our research for its better presentation and understanding by the other readers,so we considered necessary to introduce a ‘’Contributions’’ subchapter in the part 4 „Results”, where we synthesized the aspects related to the novelty, originality and added value of the work, demonstrating that it has both theoretical and practical implications.

We’ve marked the changes with pink.

It was a good learning experience.

Thank you for your time. 

Response to Reviewer 3 Comments

Point 1. The manuscript has been improved.  The fundamental problems like novelty and scientific value are still there, although the problems have been addressed to some extend in the revised version.  The rest parts are fine.

Response 1:

Line 72-79: The ammendemnts  to the Inroduction

In sum, the analysed literature sources allow us to conclude, that there is a big attention to collaborative models of smart sustainable city. It relates to both aspects: 1) a way to balance three E’s (economy, equity and environment) of sustainability by agreement of the priorities among the certain community bodies and groups), and 2) actually making a city smart through the use of edging knowledge, technologies and communication means to solve the different aspects of the city sustainability. It is quite clear, that knowledge-based structures possess the biggest portion of necessary expertise to foster both sustainability and smartness in a community; involving them in the adaptive networking would increase their potential collaborative power.   

Line 72-79: The ammendemnts  to the Results (part 4)

4.3. Contributions

The smart cities concept exploited in our research and based on multi-stakeholder collaboration and networking with emphasis on interconnections of different aspects of urban life, is proving the importance of a holistic view on sustainability and smart-city (in a mixed format), although is often debated in the literature of the last years, and the new challenges regarding these approaches are still arising. The contribution of our research to the completion of knowledge in this field has both theoretical and managerial implications

Theoretical implications

Our research identifies and formulates in a novel and original way, that a concept of a sustainable city as a smart-organization can be considered as a complex and adaptive system, which is functioning on a base of permanent adaptations of human actions towards desired sustainable state, resulting from a city’s behaviour as a whole system, consisting of natural, economic, social and other involved components.

 The adaptability here is interpreted as dynamic learning and organizational processes in networks of Groups of Competences, which are through permanent structural improvements accumulate necessary knowledge, serving to or/and directly solving the arising issues of smart sustainable city. The Groups of Competences in this way generate the additional economic, social and environmental values within the adaptive smart sustainable city model and enhance its smart entrepreneurship. Thus, the suggested new paradigm confers our research added value to holistic philosophy and to organizational and learning designs of smart sustainable city.

The holistic philosophy of smart sustainable city was shaped through using the complex adaptive system theory (CAS) as a dynamic frame for understanding a network of agents that collaboratively could form a smart-city model and respond to sustainability goals. The sustainable move will depend on how much the different components of smart city are driven by, can adapt and contribute to common sustainability goals.

The notion of organizational design of smart sustainable city was developed through suggestion and further interpretations of Groups of Competences as the contouring and centring nodes of collaborative networking, generating its format of a system, complexity, adaptability, and larger resulting power of working together, what reflect the features of CAS model.

Enhancing the learning concept of smart city was provided by adaptive heterogenic and interdisciplinary networking of the Groups of Competence connected to sustainability goals. Their members are from different fields, grouped by their competences in the main dimensions of development of the smart sustainable city. They work together in a high interconnectivity, having common goals towards excellence in innovative and smart solutions through the diffusion of knowledge.  

         The model was developed and technically tested by the means of SNA on a base of innovation Cluster and resulted in defined Groups of Competences aimed at addressing innovative and sustainable solutions through applied R&D, which underpin ‘Smart’ components of cities (Smart Economy, Smart Technology, Smart Infrastructure, Smart Work and Living, Smart Mobility, Smart Citizen, Cleaner Environment, Smart Health Care).            

b.Managerial implications

Participating in collaborative innovative networks within the smart-city frame, serve both the community sustainability goals and, through acceptable innovations, better survival in market, if this is a business type of partner. Thus, on a higher level, we see the benefits of diversity and complementarity of the network together with the intense connections between the activities involved. Thereby, the results are of a real help to all those involved in solving the practical problems of smart cities: local authorities, business managers, public institutions, non-governmental organizations that are considering innovation in smart cities through the Smart Cities Collaborative Network; the conceptual model allows its replication under different conditions, by testing, validating and co-creating innovations in smart cities. Understanding the smart city collaboration networks and the relationships within its behaviour can help stakeholders to decide what they need and thus, stimulate their business and innovation goals with the help of the Smart Cities Collaborative Network. These collaborative forms will have the effect of intensifying, diversifying and increasing the quality of innovations in the smart city. ‘’